# Recycling of Chrome-Tanned Leather and Its Utilization as Polymeric Materials and in Polymer-Based Composites: A Review

**DOI:** 10.3390/polym13030429

**Published:** 2021-01-29

**Authors:** Mariafederica Parisi, Alessandro Nanni, Martino Colonna

**Affiliations:** Department of Civil, Chemical, Environmental and Materials Engineering, University of Bologna, 40131 Bologna, Italy; alessandro.nanni11@gmail.com

**Keywords:** biocomposites, collagen, leather, recycling, sustainability, mechanical reinforcement, fillers

## Abstract

Tanneries generate large amounts of solid and liquid wastes, which contain harmful chemical compounds in the environment, such as chromium, that is used in the tanning process. Until now, they have been almost completely dumped in landfills. Thus, finding eco-sustainable and innovative alternatives for the management and disposal of these wastes is becoming a huge challenge for tanneries and researchers around the world. In particular, the scientific and industrial communities have started using wastes to produce new materials exploiting the characteristics of leather, which are strongly connected with the macromolecular structure of its main component, collagen. None of the reviews on leather waste management actually present in the scientific literature report in detail the use of leather to make composite materials and the mechanical properties of the materials obtained, which are of fundamental importance for an effective industrial exploitation of leather scraps. This comprehensive review reports for the first time the state of the art of the strategies related to the recovery and valorization of both hydrolyzed collagen and leather waste for the realization of composite materials, reporting in detail the properties and the industrial applications of the materials obtained. In the conclusion section, the authors provide practical implications for industry in relation to sustainability and identify research gaps that can guide future authors and industries in their work.

## 1. Aim and Structure of the Review

The present review is set within the context of leather manufacturing and in particular the problems connected with its production and recycle. Indeed, leather manufacturing is one of the most ancient and widespread industrial activities in the world, in which a material (leather), mainly composed of a natural macromolecule (collagen), is chemically modified to obtain durable goods. The global leather industry produces about 1.7 billion m^2^ of leather, with an estimated market value of about 34 billion euro [1]. Currently, the world’s biggest leather producers are located in Asia, with China being the leader of all prominent countries in the leather industry, followed by India and Hong Kong. Among the EU countries, Italy is the leader in this sector, followed by France, while Germany, as the biggest importer of the EU, imports mostly from Turkey, China, and India [2,3].

Finished leather is obtained by treating animal skins and hides with chemicals (tanning agents) in order to modify the macromolecular structure of collagen and make them suitable for use as clothing, footwear, handbags, furniture, tools, and sports equipment [3]. This process involves three basic steps: pre-tanning, tanning, and post-tanning (Scheme 1), which are followed by a last finishing step before commercialization [4,5]. The series of operation involved in leather production require a huge amount of water and other chemical substances, discharging solid and liquid wastes into the environment [6]. One of the biggest problems regarding the management of tanneries wastes is the presence of chromium in tanned leather scraps and shavings. Thus, finding a sustainable solution for the disposal of these solid wastes is becoming a relevant challenge for tanners and researchers [7]. The scientific and industrial communities have therefore started using wastes to produce new materials exploiting the characteristics of leather, which are strongly connected with the macromolecular structure of its main component, collagen. In particular, the use of collagen after hydrolysis to make polymer-based materials and the direct use of chrome-tanned wastes as reinforcing agent in polymer-based composites have been widely investigated in the last years. However, the industrial readiness of the current developments is generally quite far from full industrial application. Some research gaps are still present in the scientific literature. For example, most of the papers on polymer-based composites do not report the effect of the process on the Cr (VI) content. Moreover, the standards used for measuring the chromium content cannot be easily applied to polymer-based composites.

Eight previous reviews deal with leather recycling and disposal. However, six of them do not report industrial applications in the field of polymer-based composites and the only two [7,8] that report this important and promising sector do not report in detail the mechanical properties of the final materials, which are of fundamental importance in view of an effective industrial exploitation of the materials obtained. Therefore, the objective of the present review is to report all of the most recent advances of the research in the field of the recycling of chrome-tanned leather wastes, which involves the use of leather macromolecular constituents and/or the use of other biopolymers to make composite materials, with the final aim to propose practical implications for the industry in relation to sustainability and to identify research gaps that can guide future authors and companies in their work.

The structure of the present review involves a first chapter that contextualizes the problems connected with leather disposal and reports the effect of leather on the environment. In the following chapter, the main waste management options are analyzed and discussed. In Section 4, the routes for the recycling of collagen hydrolysate and its industrial applications are reviewed, mainly focusing on the exploitation of the macromolecular structure of collagen. In Section 5, the direct use of leather as reinforcing agent is reported, divided by the class of polymer matrix used (thermoplastic, thermosetting, and vulcanized rubber). Particular importance is given to biopolymer matrixes (biodegradable and/or obtained from renewable resources). In Section 6, other applications (e.g., in asphalts and cements) of leather in composite materials are reviewed. In the conclusion section, practical implications for the business and for the scientific community are provided in order to highlight the gaps of the present works and to suggest new fields of research in view of the industrial exploitation of the use of leather scraps in polymers. 

The search of the scientific literature was performed using Scopus, Google Scholar, and Web of Science databases, while the patent search was conducted with Orbit. The keywords used are those reported at the beginning of this paper. 

The topics and subtopics of the papers analyzed for the present review are reported in Table 1. Five patents were also reviewed. In particular, three of them deal with chemical/enzymatic treatments, one with the use of protein fraction in tanning processes, and one with the production of leatherette materials with different biodegradable matrices.

## 2. Raw Leather Materials and Solid Waste Disposal Issues

### 2.1. Raw Materials, Reagents, and Emission Factors

The raw materials in the leather industry are raw hide or skin. Bovine skin consists mostly of water and proteins, such as collagen (29%), keratin (2%), and elastin (0.3%), and as minor components fats and other inorganic substances [9].

Collagen is a unique protein, characterized by an uncommon hydrothermal stability even in its native state [10]; it represents one of the most abundant structural proteins in all animals and it accounts for one third of the total protein [11]. There are 28 different types of collagen composed of at least 46 different polypeptide chains [12,13]. The most abundant collagen is the type I, which is present within skin, tendons, organs, ligaments, and bones. From a structural point of view, the repeating unit of collagen is formed by three parallel polypeptide strands in a left-handed α-helical conformation coil about each other with a one-residue stagger to form a right-handed triple helix (Figure 1). In collagen type I, the three stands are composed by two α_1_ identical polypeptide chains and one slightly different α_2_ chain. In each single alpha chain, it is recognizable as a repeating unit having the following sequence: (Gly-X-Y)_n_, where Gly is glycine, which represents around one third of collagen, and X and Y can be any amino acids (Figure 2). Generally, the amino acids in the X and Y position of collagen are proline (Pro, 28%) and hydroxyproline (Hyp, 38%), respectively [11]. In particular, Gly-Pro-Hyp is the most common sequence and represents 10.5% of all possible triplets [14]. Proline is always adjacent to glycine for steric hindrance reasons: in fact, glycine is the smallest amino acid, and it needs to create free space to bulky proline in order to form the compact and tied helical structure of collagen, in which proline molecules represent the kinks. On the other hand, hydroxyproline (or hydroxylysine), formed in the endoplasmic reticulum with the hydroxylation of proline (or lysine) by ascorbic acid, has the function of stabilizing the triple helix by maximizing hydrogen bonds between individual alpha chains [11]. Other amino acids often present within collagen in the X and Y position are alanine, glutamic acid, aspartic acid, serine, lysine, and leucine [15].

Collagen aggregates into microfibrils and then into fibrils. These structures are bonded together to create fibers organized in a complex network (tropocollagen) [3,16]. The hydroxyproline ring, which is on the outer part of the collagen helix, constitutes an aggregation point for polar molecules, such as water, that bond to the hydroxyl prolyl group via hydrogen bonding [3,17], creating a water layer.

Tanning agents used to preserve skin/hide (mostly as chromium (III) sulphate) [3] interact both with the triple helix of collagen and its supramolecular water layer [3,18] by different interactions, including covalent, hydrogen, and Wan der Waals bonding, involving both the acid and basic group of the amino acids (Figure 3) [3,19]. Therefore, the cross-linking between the tanning agents and the triple helix of the collagen is responsible for some of leather’s peculiarity and, above all, for its stability [3,20].

Along with chromium salts, other chemicals are used in the leather industry, such as lime, sodium sulphite, ammonium salts, sulfuric acid, and, in some cases, vegetable tanning agents [21,22]. It has been reported that about 1000 kg of salted hides, which generate 600 kg of solid wastes, are necessary to produce 200 kg of finished leather [23]. Moreover, the use of the above-mentioned chemicals to convert raw skin into finished leather leads to other liquid wastes. In particular, the processing of 1 metric ton of skin provides 50 m^3^ of wastewater [24]. Worldwide, tanneries produce liquid and solid wastes with emission of 1470 kTon of COD (chemical oxygen demand), 619 KTon of BOD (biological oxygen demand), 920 kTon of suspended solids, 30 kTon of chromium, 60 kTon of sulphur, and 3000 kTon of solid wastes [7].

### 2.2. Solid Waste and Disposal Issues

The enormous amount of solid waste generated by the tannery consists mostly of chrome-tanned leather shavings (CTLSs) and trimmings but also of fleshings, splits, buffing dust, and hair (Table 2) [7,21].

Chrome wastes are unavoidable and pose a serious threat to the environment, due to the presence of chromium [7]. Hence, in the last decades, the disposal of these solid wastes has represented a big deal for tanneries around the world. For several years, solid waste has been discharged into landfill. This option is not sustainable anymore because it involves huge environmental problems [7], such as contamination of soil and water, increasing levels of global warming, and makes land no longer suitable for its use due to bioaccumulation of pollutants [25] (Scheme 2) [5].

Almost 30% of total proteinous waste generated by tanneries consists of CTLSs, which are small pieces of chrome-tanned leather produced during the shaving process [7,26]. The physical and chemical features of this fibrous material are shown in Table 3 [7,27].

A major problem concerning the disposal of CTLSs is the potential oxidation of trivalent chromium salts into hexavalent chromium salts, which are very soluble with a carcinogenic effect [28]. There are many factors that affect the formation of these dangerous salts, such as:The combined effect of unsaturated fatty acids present in fat-liquoring agents and photo-ageing with UV light or thermal ageing (e.g., dry heating over 353 K) [29].The combined effect of relative humidity and temperature in the storage of fatty-liquored leather.The presence of an alkaline medium used, for example, in footwear production [29].

In the absence of any particular conditions in air, chromium (III) oxidation occurs according to the following reactions [28]:2Cr_2_O_3_ + 8OH^−^ + 3O_2_

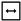
 4CrO_4_^−^ + 4H_2_O in alcaline medium (1)2Cr_2_O_3_ + 2H_2_O + 3O_2_

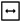
 2Cr_2_O_7_^−^ + 4H^+^ in acid medium (2)

These two reactions can take place spontaneously because of their negative Gibb’s energy and in a wide range of pH [29], and moreover, they could be catalyzed in soils by the presence of other metals, such as cerium and manganese [30]. Thus, in order to limit any degradative process after landfilling that could result in dangerous or toxic substances, it would be preferable to adopt more sustainable disposal systems than landfilling. Recently, researchers around the world have focused on the study of new safe disposal methods, which would enable the separation and recovery of chromium and the reuse of waste in various industrial fields [7].

## 3. Waste Management Methodologies

In the last few years, several treatment techniques have been developed by scientists and they can be briefly classified into the following classes (Scheme 3) [7]:Direct use to make composites.Utilization of recovered collagen from hydrolysis.Thermal treatment process.

The first two options exploit the properties of leather and collagen by the direct recycling of leather wastes for the production of new composite materials and by the re-use of collagen, previously separated from chrome by a chemical process. Energy recovery has also been exploited through different thermal treatments, such as pyrolysis and incineration.

### 3.1. Thermal Treatment Process of Chromium-Containing Solid Wastes

Thermal processes, such as incineration, pyrolysis, and gasification, are used for energy recovery. Incineration represents the easiest option since it enables the production of energy along with the consumption of chrome wastes [7]. Unfortunately, this process needs particular attention because of possible environmental damage, such as release of the toxic hexavalent chromium, and the production of halogenated organic compound and poly-aromatic hydrocarbons [31]. Moreover, this approach is the less favorable from an environmental point of view within the three approaches since it does not exploit all the positive mechanical characteristics of leather and for the increase of greenhouse gas emissions. Nevertheless, since this waste contains more than twice as much energy as coal, its conversion into useful energy is used for the energy requirements of tanneries for the processing of hides and skins [7].

Pyrolysis is another option for the recovery of solid wastes that allows several different products to be obtained, such as gas, oil, inorganic compounds, and carbonaceous residues. Gas and oil can be used as fuel or as starting materials for the synthesis of new chemicals [7], while carbonaceous residues are suitable for the production of activated carbon [32] or in addition, thanks to their high content of chromium, they can be used as pigment for ceramic materials [33]. Absorption by activated carbon is a well-known and widely used technology for the treatment of wastewater and soils and for the removal of oils, greases, and various solvents. In the last few years, there has been a growing interest in the conversion of leather shavings into activated carbons by controlled pyrolysis under an N_2_ atmosphere followed by activation with a CO_2_ stream [7]. This process allows a material with a high specific surface area and a high removal capacity of organic substances from aqueous medium to be obtained [32,34].

The gasification process represents the best compromise between power generation and disposal of wastes. This process converts all the organic matter into combustible gas, including carbon monoxide, hydrogen, and methane, which can be used as fuel for the generation of electricity and heat. The gasification process used to produce syngas results in the recovery of almost 70% of the intrinsic energy of tannery wastes [7].

### 3.2. Chemical-Enzymatic Treatments for Collagen Recovery and Chromium Removal

The separation and removal of chromium from leather shavings (de-chroming) is performed through a hydrolysis process that can be made both in acid and alkaline medium or using enzymes. The processes enable collagen to be effectively isolated from chromium as protein hydrolysate (gelatine) with a high degree of purity for use in various secondary applications.

The use of acids and complexing agents for the removal of chromium from leather scraps and shavings has been widely investigated. In particular, treatments with sulfuric [35,36] and oxalic acid enable the separation of chromium from the collagen-chromium complex, transforming chromium into a soluble salt or complex, separable from solids by filtration [35]. However, the removal of chromium from the solution is difficult, because of its strong complexation [8]. Recently, de-chroming with sulfuric acid at low temperature has been attempted. This process ensures a more efficient removal of chromium while preserving the leather structure [37].

The use of both organic and inorganic acids, such as formic, phosphoric, and nitric acid, has also been proposed for the hydrolysis of the commercially available hydrolysis product of chrome shavings Hykol-E^®^ in order to obtain a low-molecular-mass product to be used as plant biostimulator [38].

Alkaline hydrolysis has been the most investigated process, due to the higher cost of acid treatments [8,39]. Several processes using CaO at different temperatures [40,41] or combined with other substances, such as CaCl_2_ and NaOH, in sequential steps have been proposed [42]. It has been found that it is possible to obtain a hydrolysate containing about 80% of protein and less than 2 ppm of chromium [8,37,38]. Hydrolysis with strong alkali followed by acid extraction has been proposed too. Use of alkali, such as sodium hydroxide, calcium hydroxide, and others, involves the separation of chromium from the collagen complex and its precipitation as chromium hydroxide [43]. The protein hydrolysate is separable by filtration, while the chromium hydroxide is solubilized with subsequent acid extractions and then removed as soluble salt by filtration [8].

Oxidative extraction with peroxides was attempted, but it was immediately abandoned due to the high costs of peroxides [8]. Various alkaline reagents and enzymes were tested and compared; calcium hydroxide appeared to be the most effective in the de-chroming process [44]. Recently, a process based on the use of potassium tartrate in alkaline medium at room temperature has been proposed to remove up to 95% of chromium without degradation or digestion of the waste source [45,46].

Several studies have been conducted on the use of enzymes for the removal of chromium from leather wastes. One of the major problems concerning this process is its technical complexity and the efficient separation of chromium [8]. The use of different enzymes obtained by solid-state fermentation, such as *Paecilomyces lilacinus*, *Aspergillus carbonarium*, and *Pseudomonas aeruginosa*, has been proposed [47,48] with approximately a 70% yield of hydrolysate. Recently, a de-chroming process using commercial proteolytic enzymes at moderate temperatures has been developed [49,50]; the reaction takes place in a pH range between 8.3 and 10.5, hence preventing the release of chromium in the solution [8]. This two-stage process firstly involves the use of alkali solution at a moderate temperature and then the addition of the enzymes [8]. It has been observed that with 5–6% alkali, such as MgO, combined with other substances (e.g., Ca(OH)_2_, NaOH, and Na_2_CO_3_) and 1% of enzyme at a temperature above 333.5 K, it is possible to solubilize approximately 80% of the shavings [34,51,52,53]. However, these conditions are not suitable for finished leather [8]. Further studies have suggested several optimizations of this process in order to obtain a better quality protein fraction with a higher yield and minimal content of chromium [8]. Pilot plant trials for most of these processes have been validated, confirming their reproducibility on a large scale [54,55,56]. Moreover, the cost analysis also indicates that these bi-stage processes are economically viable [55,57]. In particular, a modified process using low-molecular amines has been reported; it allows a protein fraction with an 89% yield along with a filtered cake rich in chromium oxide to be obtained [34,58]. This process has been industrialized because of its easier processing. Several other studies have been conducted in this field, in general involving multi-stage hydrolysis resulting in two or more fractions [8].

## 4. Recycling of Collagen Hydrolysate and Its Industrial Applications

Recovered protein hydrolysate can find application in various fields, such as in leather manufacturing, polymers, adhesives, agriculture, animal feed, cosmetic industry, and pharmaceutical applications [8].

It has been observed that the recovered protein fraction can contribute to the tanning process, increasing the chromium oxide content and improving the leather properties [35].

Leather treated with protein hydrolysate shows a better feel, improved homogeneity, along with glossier and more brilliant colors [35]. The use of collagen hydrolysate in fatliquors, re-tanning agents, and other additives can contribute to improving the physical properties of leather [59,60,61,62]. It has been reported that leathers treated with protein hydrolysate show an increasing level of tear and tensile strength (up to 30%) with a 10% decrease in the elongation at break of the material [60].

Thermo-reversible and thermo-irreversible gels can be obtained from the reaction between gelatine and glutaraldehyde. The first ones can be employed in the production of glues, while the irreversible gels can find their application in the encapsulation techniques [62]. Collagen hydrolysate can also be used for the preparation of hydrogels, after reaction with dialdehyde starch and glycerol for the production of packaging materials with controlled release of active substances for food, cosmetic, and pharmaceutical applications [61].

Another important application of the alkaline hydrolysate concerns the adhesives formulations [63]. It has been reported that the addition of collagen hydrolysate to urea-formaldehyde and phenol-formaldehyde resin adhesives improves the binding properties, reducing levels of free formaldehyde in the cured resin [64]. Gelatine obtained from chromium wastes modified with glutaraldehyde glyoxal and carbodiimide can be used in the synthesis of adhesive too [65]. Recently, new polycondensation adhesives based on collagen hydrolysate have been developed for woodworking applications [7].

Coagulants for natural rubber can be prepared from protein hydrolysate obtained from alkaline hydrolysis [66]. Gelatine can be used as a filler for isoprene and butadiene-acrylonitrile rubber, improving thee aging resistance and microbiological degradation properties [7]. Collagen hydrolysate with low molecular mass may be used in surfactants after their acylation or quaternation [38,67,68].

The biocomposite layer of silica obtained from the coatings of silica sols mixed with protein hydrolysate has been prepared. The resulting products have shown good biodegradation capacities along with good mechanical properties [69]. The synthesis of a scale inhibitor based on chemically modified collagen with rich carboxyls has also been reported, which have a complexation function to Ca^2+^, showing good results in calcium carbonate-scale inhibition [70]. Moreover, collagen hydrolysate has also been employed for the synthesis of composite sheets based on polyvinylpyrrolidone (PVP) suitable for footwear or clothing applications [71].

Gelatine is used both in cosmetic and pharmaceutical applications as a microencapsulating agent to encapsulate drugs, essential oils, fragrances, and other substances, with no significative differences compared to industrial gelatines [72]. A synthetic process using an encapsulating agent based on gelatine and chitosan has also been reported [73].

Condensation products of mono e dicarboxylic acids can be prepared from gelatine. These products have shown a remarkable similarity to detergents that are commercially available [8].

Hydrolyzed collagen is an important renewable resource for the preparation of biodegradable plastics [8]. Biodegradable protective films for automotive applications have been prepared by the reaction between collagen and diglycidyl ethers of bisphenol A [74]. Gelatine treated with transglutaminase and glycerol as a plasticizer can be used in the production of films suitable for food and packaging applications [75]. It has been reported that the addition of polyvinyl alcohol improves the mechanical properties of these films. Most of these samples have also shown biodegradable properties [8,76]. Water-soluble films suitable for agricultural packaging can be obtained by adding glutaraldehyde during the alkaline–enzymatic hydrolysis process [77]. Water-soluble films based on PVA (polyvinyl alcohol) and sugar cane bagasse have been prepared too [78]. The resulting products can be used for the production of biodegradable materials that can release nitrogen, thus acting as fertilizer [79,80,81]. Modification of collagen with epichlorohydrin and low-density polyethylene has been proposed for the production of thermoplastic materials suitable for agricultural packaging and application [82,83]. Biodegradable free-solvent epoxy films have been obtained by adding hydrolyzed collagen during the cross-linking reaction of an epoxy resin [7].

Recently, a covalent cross-linking reaction of collagen hydrolysate with cyanuric chloride has been proposed [84]. The resulting material showed better resistance to heat and enzyme degradation, making it potentially suitable for biomedical uses.

Hybrid films based on collagen extracted from leather waste with acetic acid and mixed with starch and soy protein have been prepared [85]. The resulting products showed good mechanical properties and thermal stability, making them suitable for biomedical applications.

It has been reported that collagen proteins with high molecular weight and purity can also be used to produce fibers by electro-spinning [86]. The collagen solution has been mixed with a solution of polyvinyl alcohol (PVA) containing glutaraldehyde [86] in order to impart better resistance and stability to the fibers.

Several studies have been conducted on the use of collagen hydrolysate with different types of polymeric matrix in order to produce biocomposites suitable for biomedical applications and tissue engineering. Covalent immobilization of collagen on poly(3-hydroxybutyrate-co-3-hydroxyvalerate) (PHBV) film has been used to improve its cell compatibility [87]. The collagen-modified PHBV films showed better cell adhesion and proliferation of chondrocytes than other PHVB film or modified PHBV film, suggesting that it is a promising biomaterial for cartilage tissue engineering. PHBV/collagen-based nanofibers have been produced by electrospinning to develop innovative substrates for nerve tissue engineering [88]. It has been found that aligned nanotopographies are suitable for oriented tissues, such as nerves. These materials have shown better cell proliferation than random PHBV/collagen nanofibers. Further attempts have also been made with other biopolymers. In particular, a collagen layer has been introduced in a poly l-lactic acid (PLLA) scaffold surface to obtain a tissue engineering scaffold with enhanced biocompatibility [89]. Additionally, in this case, chondrocyte proliferation was noticed in the matrix. Composite materials comprising a collagen matrix with embedded carbon nanotubes have been described [90]; these materials may have utility as scaffolds in tissue engineering, or as components of biosensors or other medical devices. 

Collagen hydrolysate recovered from chrome-tanned leather through chemical treatments has also been used as feed for anaerobic digestors to produce biogas. Microorganisms use the potassium and phosphate contained in the alkaline hydrolysate as macronutrients for their growth. In the anaerobic digestors, increasing levels of methane generation up to 30% have been observed [7,91].

The protein hydrolysate can also be used as biofertilizer or animal feed because it represents an important nitrogen source. Collagen obtained from chromium-tanned leather waste must not contain traces of chromium and other hazardous chemical agents in order to be used as animal feed or fertilizer [8]. It has been reported that alkaline and enzymatic hydrolysates with low molecular mass have been used as bio-stimulating protein additive in various fertilizer formulations [38]. Solid pre-tanned wastes and bovine hides have been used for animal feed, because of their high content in globular proteins [92]. Many of the chemicals used in tanning operations are believed to be carcinogenic for animals, especially regarding halogenated aromatic compounds, amines, and aldehydes [8]. Several nutritional tests have been conducted on proteins obtained from alkaline hydrolysis operations; it was noticed that for hydrolysate percentages of 5%, any adverse effects on animal health were not reported [41]. Other authors have reported that protein meals prepared from leather manufacturing by-products lack some important nutrients necessary for animal nutrition [34,55,93,94].

The chromium cake obtained from the hydrolysis process is usually made of chromium compounds and other various inorganic substances. This solid fraction can be used directly in leather manufacturing, as a flocculent agent in order to precipitate chromium in the tanning liquors [95]. The chromium contained in the sludge can be recycled and re-used in the tanning process after being purified through several steps of dissolution in acidic medium, precipitation and filtration [8]. It has been observed that chromium cake with low fat can be combined with hot sodium dichromate and sulfuric acid in order to obtain a tanning agent for leather processing [96].

Other routes provide the oxidation of chromium cake to produce high-quality chromate and dichromate [97]. The presence of calcium in this solid fraction can lead to calcium dichromate during the oxidation process, which can be converted into sodium dichromate using sodium carbonate. This product is one of the most important starting materials for the preparation of other chromium salts.

Another significant way to recycle chromium from leather wastes involves treatments with enzymes and other alkaline substances (CaO, NaOH, MgO) to produce pigments, such as chrome-tin pink and cobalt-chromite green, whose formulas are CaSnSiO_5_·xCr_2_O_3_ and Cr_2_CoO_4_, respectively [98]. Other pigments may be prepared from the chromium cake obtained by alkaline hydrolysis with calcium hydroxide prior to oxidation in an oxygen-rich atmosphere [99].

## 5. Direct Utilization of Chrome-Tanned Leather in Polymer-Based Composites

Direct applications of chrome-tanned leather mostly include the use of leather waste as a reducing agent in the tanning process, as absorbent or adsorbent material, and as a filler/reinforcing agent in composite materials.

Because of their high content in protein, leather shavings can be used for the preparation of tanning agents as reduction additives for Cr (VI) [8]. The resulting products have shown a significant masking effect ascribed to the formation of intermediate oligopeptides, thus helping the tanning process by shifting the chromium precipitation point to higher pH values [97]. The reduction of hexavalent chromium depends on different parameters, such as the amount of shavings and sulfuric acid, as well as the time and temperature of the process [100]. Several quality tests, conducted on leathers tanned with these agents, have shown that there is no significant difference with the commercially available tanning agents and moreover, the formation of oligopeptidic intermediates can lead to a finished leather with a better feel and brighter colors [101].

Many studies have been conducted on the possibility of using ground leather wastes to clean industrial soil by oils, hydrocarbons, and solvents [102]. Several pilot plants concerning the use of chrome-tanned leather shavings (CTLs) as an absorber for the removal of chloride, fat-liquorings, tanning agents, and other chemicals from wastewaters have been successfully developed [8]. It has been reported that solid tanned waste can be used as adsorbent material for the removal of heavy metals, especially hexavalent chromium, with a maximum concentration of 133 ppm of chromium and pentavalent arsenic from aqueous media with an arsenic concentration of 26 ppm [103].

The methods used to determine the Cr (VI) content in leather are all based on the measure of extractable chromium (VI) that is leached from the sample at pH 7–8 using a phosphate buffer. Since mobility, and therefore leaching, is strongly decreased inside polymer matrixes, this type of test, based on ISO 17057 norm, can give rise to results that do not reflect the real content of Cr (VI) inside the composite material. Moreover, the determination of chromium (VI) is mainly performed using colorimetric methods that can be influenced by additives present in the plastic materials. Other extraction methods of chromium from the polymer matrix (e.g., microwave digestion, which is used to measure the total chromium content) can contribute to the oxidation of chromium (III) and therefore can have an effect on the final measure. Probably for this reason, most of the papers that deal with the preparation of composites containing leather do not report the chromium (VI) content of the final composites. However, as previously reported, the Cr (III) oxidation to Cr (VI) is accelerated by temperature and most of the polymer/leather composites are obtained by melt mixing methods that occur at temperatures often exceeding 150 °C. Therefore, it is of fundamental importance, in view of industrial applications, to report the amount of Cr (VI) in the final materials. Finally, to correctly measure the Cr (VI) content, it is necessary to develop a new standard procedure for leather/polymer composites.

### 5.1. Composite Materials with Chrome-Tanned Leather (Granules, Dusts and Fibers)

The use of chrome-tanned leather for the preparation of composite materials has been widely reported in the scientific literature. The process requires the preparation of leather scraps by grinding (Figure 4 and Figure 5), followed by the addition of binders, such as thermoplastic or thermosetting polymers, elastomers, and other materials. Additives, such as plasticizers, antioxidants, fillers, and pigments, may also be present. These composite materials are generally fabricated by the means of a twin-screw extruder or internal mixers, and the obtained pellets are further processed by compression or injection molding (Figure 6). These series of operations allow the disposal of different types of waste resulting from leather processing for various applications, such as footwear, fashion accessories, automotive, and buildings [8].

#### 5.1.1. Thermoplastic Composite Materials

The intrinsic fibrous nature of leather waste allows its use as reinforcing agent in many thermoplastic composite materials [8], with the possibility of adding additives or modifying leather fibers by in situ polymerization with other polymers (e.g., methyl methacrylate) in order to improve compatibility with some thermoplastic commodities, namely polyethylene (PE), polypropylene (PP), polyvinyl chloride (PVC), and polystyrene (PS) [104].

Composites incorporating leather fibers have been prepared in order to improve polyvinyl alcohol’s (PVA’s) mechanical properties for packaging applications. The compounded materials have been prepared, showing that composites with a fiber content of 5% and an average diameter of 70 μm exhibit a superior processability along with better adhesion between the fibers and matrix and therefore better mechanical properties in terms of tensile strength, elongation, and thermal stability [105].

The use of natural wastes as reinforcing fillers/fibers within biopolymers (bio-based and/or biodegradable polymers) is increasingly gaining importance. In fact, being not edible and renewable, natural by-products can decrease the biopolymers’ price and improve their mechanical properties without affecting (in same case enhancing) their bio-based content. Moreover, this route of valorization would offer new possibilities to the agro-industrial companies, which are often in trouble with the disposal and management of their wastes [106]. Examples may regard the use of coffee wastes, wine wastes [107,108], and rice wastes [109]. Similarly, in the last years, leather wastes have started to be tested as reinforcing fillers within different biopolymers. As an example, with the aim of developing eco-biocomposites using leather waste to reduce pollution and provide an environmentally sustainable solution, composites with leather fibers based on polylactic acid (PLA) have been developed [110,111]. Leather fibers finely ground with an average diameter of 45 µm have been mixed with PLA in variable percentages ranging from 0–20%. Composites with a 10% fiber content have shown a 25% increase in the module compared to virgin PLA [110]. Other attempts using polylactic acid have been made using leather fibers chemically modified with silanes containing epoxy groups, mixed by solvent casting with PLA modified with trimethyl vinyl siloxane [111]. The chemical modifications have contributed to an enhancement in both the compatibility between PLA and leather fibers and the dispersion of the modified fibers in the PLA-modified matrix, thus obtaining better thermal and mechanical properties, such as the elongation at break and impact strength, compared to the raw PLA [111].

A process for the production of a leatherette material with leather waste has been developed [112] compounding leather with various biodegradable thermoplastic matrices, such as polyamides, PCL, PHB, PHBV, and PLLA. The scraps were reduced to 0.1–5-mm-long fibers by a mechanical process and were added in percentages from 5% to 15% to the polymer matrix [112]. The resulting materials showed a homogeneous and uniform structure, indicating a good interaction between the fibers and matrix. Thermal analyses showed that the thermal stability of the polymer is not affected after the addition of fibers up to 15% [112]. The results suggest that these materials could represent a viable alternative to leather obtained from conventional raw materials. However, it has to be considered that after the biodegradation of the polymer matrix, all the metals that are inside the composite are left in the compost. Therefore, the chromium that is present in leather can have a negative effect on the toxicity of the compost and on the life of the microorganisms present in the compost. Indeed, the ISO EN 13423 standard for industrial compostability defines a limit of 50 mg of chromium per kg of biopolymer and the amount of chromium in leather is significantly higher than this value. Moreover, the eventual presence of Cr (VI) can give rise to more problems due to its high toxicity. Therefore, only biocomposites with very low amounts of leather can be safely composted.

Composite materials formed by leather scraps and fossil-based and non-biodegradable polymers using polymers have also been described in the literature. In particular, leather trimming wastes and household garbage have been used in an LDPE/LLDPE (50:50) blend for the production of 3-mm-thick films by compression molding. It has been found that the addition of leather waste improved some of the mechanical properties of the material, such as the hardness, tensile, and flexural strength. Moreover, tear strength is increased with a different polymer ratio [113]. Fibers with a 0.5-mm particle size were compounded in varying contents from 0% to 60% with HDPE and compression molded [107]. The addition of additives, such as inorganic fillers or more flexible polymers (natural rubber and ethylene-vinyl acetate), led to an improvement in the mechanical properties of the composite, such as yield stress, tensile, and impact strength, especially for a fiber content of 10% [114]. In composite films with PVC as polymer matrix, it has been noticed that the size of the fibers plays a fundamental role on the properties of the composite [108]. By decreasing the size of leather particles, the weak points of the stress concentration in tested specimens were reduced. In addition, the coating of fibers with EVA resulted in an improvement of 30% of the mechanical properties, such as the tensile strength of the material [115].

#### 5.1.2. Thermosetting Composite Materials

Several studies have been conducted on the possibility of using crust leather to produce valuable thermosetting composites, such as fibrous sheets, boards, and additives, for building applications. In particular, fibrous composites based on polyester resin have been developed from post-consumer leather powder with sizes between 300 and 500 microns [116]. The solution containing polyester resin, PVA, and initiator and leather in percentages ranging from 0% to 60% was sandwiched between two polystyrene sheets and compression molded to the required thickness value. The obtained panel was left at room temperature for the curing of the resin. The sample with a fiber content of 30% showed a better tensile strength and a higher modulus compared to the other composites, due to the better distribution of the fibers in the matrix and thus to a better adhesion between them [116]. Leather fibers have also been used as reinforcing materials for epoxy resin laminates in combination with carbon nanoparticles [117]. The composites showed enhanced mechanical properties, such as tensile, flexural, and impact strength, along with an improved elongation at break and a higher modulus as well as electrical conductivity [117].

Other approaches concerning the use of leather waste for the production of composite materials have been reported for the preparation of construction materials. In particular, composites based on polyisocyanates and other resins have been used for the preparation of fibrous sheets grafted with hydrophilic acrylates. The use of leather has also been reported [8] for the preparation of wooden panels in which the leather fibers partially replace the wooden particles in composites made of formaldehyde-urea resin. The composites containing leather possess improved mechanical performances combined with a lower content of free formaldehyde [8].

#### 5.1.3. Rubber-Leather Composites

The rubber industry is constantly in search for fillers and additives in order to improve the durability properties of their products. The reinforcement of elastomers allows an improvement of their adhesion, increasing the tear strength and the abrasion resistance of the material. For this reason, several studies have been carried out on the recycling of solid leather waste for the production of elastomeric-based composite materials. For example, chromium-tanned leather fibers with an average size of 1 cm in length and 0.05 mm in diameter were added to natural rubber and compression molded. The material exhibited better mechanical properties, along with softness, flexibility, and thermal stability, with respect to standard rubber. In addition, the rubber-leather composite showed good biodegradable properties and breathability [118].

The recycling of natural rubber, using both untreated and neutralized leather particles, has been investigated [119] with the additional aim to promote the consumption of rubber scraps. Neutralized leather shavings added to rubber scraps have been compounded with virgin natural rubber and vulcanized by compression molding. It has been found that the use of treated leather particles permits the addition of significant amounts of rubber waste into virgin rubber latex, constituting a continuous and dispersed phase in the rubber matrix, without seriously affecting the vulcanization characteristics [119]. Moreover, neutralization of leather particles provided better properties of the vulcanized material, reducing the swelling both in organic and aqueous media [119].

Reconstituted fiber composite materials have been prepared mixing leather, cotton, and polyester fibers with a natural rubber matrix. The compounded paste has been compression molded into sheets and sun dried [120]. The resulting composite material has shown better or comparable strength properties as compared to natural rubber containing only leather fiber [120]. The results have suggested that this reconstituted material possesses adequate mechanical properties for the manufacture of good and footwear consumer products.

Composite materials have also been made using other types of rubber. In particular, acrylonitrile butadiene rubber-based composites were prepared using solid tanned leather scraps [121]. Leather waste was disintegrated into fine dust with a particle size of 0.2 mesh and neutralized. The treated leather was added in contents ranging from 0 to 10 phr to the rubber matrix and the resulting compound was vulcanized by compression molding [121]. The presence of the leather fibers influenced the mechanical properties of the composite, improving the tensile strength and the Young’s modulus at the expense of the elongation at break of the material. The produced composites showed enhancements in the ageing coefficient [121], suggesting that both treated and untreated leather possess the inherent ability to resist to thermal ageing, improving the thermal stability of the materials while reducing the cost of the final rubber matrix. Rubber-leather composites suitable for footwear functional applications, such as soles and in-shoe parts, have been prepared using short leather fibers and granules with an average length of 1 mm mixed with a styrene butadiene and acrylonitrile butadiene rubber matrix in a range of 12.5 to 300 phr [122]. The compounded materials were vulcanized by compression molding, showing, for a leather content between 10 and 20 phr, a 15% improvement in the tear resistance of the material [122]. Products obtained with higher fiber percentages still maintain acceptable mechanical properties for in-shoe applications.

Additionally, EPDM elastomers have been used to prepare composites with leather. In particular, particles with a diameter of 1000–2000 micrometers were added to an EPDM matrix in various proportions ranging from 0 to 20 phr [123]. The compounds were vulcanized and compression molded. Test results conducted on these materials showed that the thermal stability of EPDM was not affected by the presence of leather fibers [123]. The decrease in the tensile strength and elongation at break of the samples suggested that leather particles interfere with the matrix, reducing the mobility of the rubber chain [123]. The most remarkable effect due to leather fiber incorporation was the improvement in the tear strength of composites before and after thermal ageing. This behavior was ascribed to the intrinsic fibrous nature of leather capable of preventing crack growth in the material [123].

## 6. Applications of Leather Wastes in Composite with Other Materials

Applications of leather waste with other materials are reported in the literature. For example, leather shavings have been used as a possible source for natural organic semiconductors [124]. Leather dust has also been used in asphalt mixtures [125]. It has been found that the addiction of 0.3 % of leather particles in an asphalt micro-surface layer improved its engineering properties, reducing the cracking of the pavement layer caused by the presence of the leather fibers in the asphalt micro-surface [125]. Another interesting application of leather waste concerns the use of leather fibers in combination with Portland cement for the realization of paving blocks for pedestrian use [126]. The physical-mechanical analyses carried out on the resulting materials showed values within the limits provided by standards for industrial application.

## 7. Conclusions

Every year, tanneries around the world generate large quantities of solid chrome-tanned waste, which until now have usually been dumped into landfills. In this review, we reported the main developments of the processes for the re-use of tanned leather. In this conclusion chapter, we propose practical implications for industries in relation to sustainability and we identify research gaps that can guide future authors and companies in their work. Since the presence of chromium in tanned leather scraps and shavings gives rise to a wide set of environmental problems, we considered this fundamental issue in all of the paper and in particular in this conclusion chapter. For clarity, we have divided this conclusion section depending on the following recycling routes of leather waste (Scheme 4):Thermal treatments and energy recovery.Hydrolysis of leather to obtain collagen and chromium-containing mixtures.Direct utilization of leather.Preparation of leather/polymer composites.

### 7.1. Thermal Treatments

Thermal treatments for energy recovery have been widely reported in the literature. Nevertheless, even if very strict control over the operating process and emissions is performed, this process can give rise to several environmental problems. Moreover, this process is not favorable in terms of greenhouse gas emission and carbon atom efficiency and therefore other recycling or valorization methods must be preferred from an environmental perspective.

### 7.2. Hydrolysis of Leather

The simple hydrolysis of leather without the removal of chromium can be used to produce a solution that can be employed for new tanning of leather, by mixing the virgin untreated leather with the hydrolysate solution, since it increases the overall chromium content. The industrial application of this process is straightforward from an industrial point of view since it requires low investment costs and can contribute to reducing the use of chromium salts and the costs of waste management. However, the preparation of tanned leather without the addition of chromium, just using the chromium coming from the hydrolysate leather, has never been industrially achieved. The development of this type of process can contribute significantly, from an industrial point of view, to decreasing the amount of chromium salts used in tanning processes.

Collagen can be separated from chromium during the hydrolysis of leather performed in acidic and alkaline aqueous media or using enzymes. Recovered de-chromed protein hydrolysate can find application in various fields, such as in agriculture and animal feed. However, in the last few years, other more interesting applications that exploit the macromolecular structure of collagen have been developed for the production of polymeric materials and composites, which can find application in various industrial fields, such as adhesives, bioplastics, and construction. In particular, of significant industrial interest is the use of recovered collagen for applications as adhesive and binder agent after cross-linking with glutaraldehyde. Crosslinked hydrogels can also find applications in cosmetic and pharmaceutical applications as microencapsulating agents to encapsulate drugs, essential oils, fragrances, and other substances with no significative differences compared to industrial gelatines.

Several studies have been conducted on the use of collagen hydrolysate with different types of polymeric matrixes in order to produce biocomposites suitable for biomedical applications and tissue engineering. This type of material that use, for example, PHBV or PLLA as polymer matrix seems to have good potential for the production of scaffolds for bone and tissue engineering.

### 7.3. Direct Use of Leather

Leather shavings can be directly used as absorbent materials for cleaning soils or wastewater from oils and other chemicals. Several pilot plants for the use of leather waste as an absorber for the removal of chloride, fat-liquorings, tanning agents, and other chemicals from wastewaters have been successfully developed, showing that this route can be of interest for the industrial community.

### 7.4. Preparation of Leather/Polymer Composites

Direct utilization of solid tanned waste has been widely reported for the preparation of composite materials. Fibers, granules, and dusts obtained from different leather wastes can be used as filler or reinforcing agents in various polymer matrixes, combined with other additives or after chemical modifications. The resulting composites have properties that depend on the leather content and on the compatibility between the polymer matrix and leather. However, the scientific studies performed have not yet fully addressed the effect of adhesion between the matrix and leather and the effect of compatibilizer on the enhancement of composite properties. For this reason, the authors of this review strongly encourage scientists to address this research gap, which can have a significant effect on industrial applications, especially in the case of non-polar polymers, such as polyethylene and polypropylene. Another topic that needs to be studied in more detail by scientists is the effect of the particle size and processing conditions on the composite properties and of the effect of leather on surface properties, such as the coefficient of friction and wettability.

Several studies have been made on the use of biodegradable polymer matrixes (e.g., PHA and PLA). However, it has to be considered that after the biodegradation of the polymer matrix, all the metals that are inside the composite are left in the compost. Therefore, the chromium that is present in leather can have a negative effect on the toxicity of the compost and on the life of microorganisms present in the compost. Indeed, the ISO EN 13423 standard for industrial compostability defines a limit of 50 mg of chromium per kg of biopolymer. Moreover, the eventual presence of Cr (VI) can give rise to more issues due to its high toxicity. For this reason, the most promising class of biopolymers to be used in the preparation of composites with leather are polymers derived from renewable resources that can be efficiently recycled and therefore do not release chromium in the environment.

Of particular interest for industrial applications are rubber/leather composites that have already found some applications, for example, in the field of footwear, fashion accessories, and automotive. The industrial readiness of the industrial processes is in some cases close to industrialization. However, more work is needed to expand the process to a wider set of industrial applications with cost-effective solutions, making recycling and valorization of solid tanned waste not only important to reducing environmental pollution but also a real economic benefit for tanneries.

One of the main limitations that needs to be solved in the near future regards the measure of Cr (VI) in composite materials. Indeed, the methods used to determine the Cr (VI) content in leather are all based on the measure of extractable chromium (VI) after leaching. Since mobility, and therefore leaching, is strongly decreased inside polymer matrixes, this type of test can give rise to results that do not reflect the real content of Cr (VI) inside the composite material. Moreover, the chromium oxidation to Cr (VI) is accelerated by temperature and most of the polymer/leather composites are obtained by melt mixing methods that occur at temperatures often exceeding 150 °C and an increase of the Cr (VI) content during composite preparation cannot be neglected. Therefore, it is of fundamental importance in view of an industrial exploitation of these composite materials to develop a precise and robust method to correctly measure the Cr (VI) content in leather/polymer composites.

## Data Availability

The data presented in this study are available on request from the corresponding author.

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
