# Peer review of "Recycling of Chrome-Tanned Leather and Its Utilization as Polymeric Materials and in Polymer-Based Composites: A Review"

_polymers, 2021, doi:10.3390/polym13030429_

Round 1

Reviewer 1 Report

This review paper reports the state of the art of the strategies regarding the recovery and valorization of such solid and liquid wastes from tanneries. More specifically, it focusses on the recycling process of both hydrolyzed collagen and leather waste for the realization of composite materials using polymers derived from renewable sources.

In general, the study has a merit but there are some issues that have been addresses before publication:

Title

The title of the paper reflects the information reported

Abstract

The abstract is good, but it must be more descriptive about the contribution of this research

Language

The writing language and style of the paper is, generally, good

Introduction section /

The introduction section, is quiet good and describes the research field under study

Methodologies

This section describes in details the methodologies used, but it must be enriched with the insertion of tables describing the “statistics” of the papers reviewed (eg. How many papers used the method X, the method Y etc.).

Also, the authors should refer to the surveying methodology, namely, the keywords and the scientific journals search engines used

Conclusions

The conclusions section must be empowered with more information about the contribution of the present study to the body of knowledge

Reviewer 2 Report

Authors aim to provide a review of the recycling of chrome-tanned leather and its utilization as polymeric materials and in polymer-based composites. The presented review is extensive, but it lacks the clear elaboration of its contribution to the field, research goals, methodology, and discussion. 

The current version of the introduction is too long, and it should be divided into two chapters: Introduction and Literature review. 

The introduction section needs to be rewritten with much better motivation and providing the context for this work. It should include Contextualization,   Importance/Relevance of the Theme, Research Gap, Objectives, and the Description of the structure of the Paper.

Your review should have two main goals: to propose practical implications for the business purpose in relation to sustainability and to identify research gaps that would guide future authors in their work. 

Authors should reconsider to explain the section about the scientific contribution in the introduction, as well as in the conclusion part of the paper, with the structured comparison of the current research with previous research. The text can be one paragraph long, but it should contain the most important studies. Please, find similar reviews and elaborate on what is new in your review, and how you contribute to the field with your research. 

The methodology section should be added, in which authors should elaborate on how they have selected the studies for their review. It seems that the review is not systematic, but even for non-systematic reviews, there should be a methodology section, that should clearly elaborate on the selection of studies, or the used approach. 

In the last section, please focus on “Discussion, Implication, and Conclusion”, and include the following issues: Summary of the research, Managerial and Academic Implications, Limitations of the paper, and Future Studies and Recommendations. In this section, it would be useful to add some kind of summary table of figure that could be instructive to the reader in order to fast grasp the main conclusions. 
